# Marijuana use and coronary artery disease in young adults

**Jeremy R. Burt** [1]*, **Ali M. Agha**[2], **Basel Yacoub** [1], **Aryan Zahergivar** [1], **Julie Pepe**[3]

**1** Department of Radiology & Radiological Sciences, Medical University of South Carolina, Charleston, South Carolina, United States of America, **2** Department of Internal Medicine, McGovern Medical School at University of Texas - Houston, Houston, Texas, United States of America, **3** Translational Research Institute, AdventHealth Orlando, Orlando, Florida, United States of America

* burtje@musc.edu

**Data Availability Statement:** All relevant data are within the manuscript.

**Funding:** The authors received no specific funding for this work.

## Abstract

### Background

Marijuana is the most popular drug of abuse in the United States. The association between its use and coronary artery disease has not yet been fully elucidated. This study aims to determine the frequency of coronary artery disease among young to middle aged adults presenting with chest pain who currently use marijuana as compared to nonusers.

### Methods

In this retrospective study, 1,420 patients with chest pain or angina equivalent were studied. Only men between 18 and 40 years and women between 18 and 50 years of age without history of cardiac disease were included. All patients were queried about current or prior cannabis use and underwent coronary CT angiography. Each coronary artery on coronary CT angiography was assessed based on the CAD-RADS reporting system.

### Results

A total of 146 (10.3%) out of 1,420 patients with chest pain were identified as marijuana users. Only 6.8% of the 146 marijuana users had evidence of coronary artery disease on coronary CT angiography. In comparison, the rate was 15.0% among the 1,274 marijuana nonusers (p = 0.008). After accounting for other cardiac risk factors in a multivariate analysis, the negative association between marijuana use and coronary artery disease on coronary CT angiography diminished (p = 0.12, 95% CI 0.299–1.15). A majority of marijuana users were younger than nonusers and had a lower frequency of hypertension and diabetes than nonusers. There was no statistical difference in lipid panel values between the two groups. Only 2 out of 10 marijuana users with coronary artery disease on coronary CT angiography had hemodynamically significant stenosis.

### Conclusion

Among younger patients being evaluated for chest pain, self-reported cannabis use conferred no additional risk of coronary artery disease as detected on coronary CT angiography.

**Competing interests:** The authors have declared that no competing interests exist.

## Introduction

Marijuana is the most popular drug of abuse in the United States and the prevalence of its use has been on the rise among both adolescents and adults [1]. Inhalation or ingestion of cannabis leads to a myriad of physiological changes, most commonly tachycardia modulated by increased sympathetic activity. It may also cause increased systolic and diastolic blood pressures in supine individuals, orthostatic hypotension and orthostatic pre-syncope. A decreased left ventricular ejection fraction, end diastolic volume and stroke volume have also been observed in subjects using marijuana. Cardiac output remains unaffected as stroke volume and heart rate change in opposite directions [2].

Smoking of marijuana has been reported as a trigger of acute myocardial infarction (MI) in 3.2% of 3,882 patients interviewed by Mittleman et al. Their study also reported that the risk of onset of MI is elevated 4.8 times in the first hour after marijuana use and rapidly decreases thereafter [3], which may be explained by marijuana induced coronary vasospasm [4]. Despite similar findings reported elsewhere [2, 5], evidence on the association of marijuana with the development of coronary artery atherosclerotic disease has not yet been fully elucidated. A systematic review of 24 articles by Ravi et al. assessed the effect of marijuana on cardiovascular risk factors and outcomes and reported that evidence in published literature is insufficient to make any conclusions [6]. This lack of knowledge is alarming as CAD is a major cause of morbidity and mortality in developed countries. Currently, the well-established risk factors of CAD include diabetes, hypertension, high total cholesterol, low HDL level, smoking tobacco and advanced age [7].

This retrospective study aims to determine the frequency of CAD among young adults presenting with chest pain who currently use marijuana as compared to nonusers. Only young adults were assessed in this study because of the high prevalence of marijuana use among this subpopulation [8].

## Materials and methods

### Patients

The study protocol was approved by the Institutional Review Board (IRB) at AdventHealth Orlando, Florida, USA. It was performed in compliance with the Health Insurance Portability and Accountability Act. The need for written informed consent was waived by IRB because of the study's retrospective nature and involvement of no more than minimal risk. All patients were from a single, large multihospital institution. A Montage® search (Montage Healthcare Solutions, Philadelphia, PA, USA) of the radiology imaging database was performed for patients who had undergone coronary CT angiography for evaluation of undiagnosed chest pain or anginal equivalent between January 1, 2016, and January 15, 2017. The search was limited to patients between the ages of 18 and 40 for men, and 18 and 50 for women. This age group was selected based on results from previous research in our lab using data from this patient cohort [9]. Patients with the following criteria were excluded from the study: 1) known coronary artery disease, 2) lack of demographic information on electronic health records (EHR), or 3) lack of social history on EHR. Marijuana use was defined as either self-reported marijuana use or positive urine toxicology screening for marijuana during this encounter or a previous encounter within 1 year of the ED visit.

Other information also collected from EHR included: body mass index, diabetes mellitus, defined as self-reported diabetes mellitus or HbA1c > 6.5%; hypertension, defined as self-reported hypertension, prior documented systolic blood pressure >140 mmHg or diastolic blood pressure > 90 mmHg; self-reported current or prior tobacco smoking; cocaine use was

defined as either recent, within the past week, or remote, more than one week prior; lipid profile values obtained within 1 year of the coronary computed tomography angiography (CCTA) were also collected. "Premature CAD" was defined as coronary atherosclerosis on CCTA in men ≤ 40 years and women ≤ 50 years of age.

## Coronary CT angiograms

A majority of patients were imaged using Siemens SOMATOM Definition Flash (Erlangen, Germany) or Philips Ingenuity (Amsterdam, Netherlands) 128-row CT scanners. Contrast doses ranged between 80 and 150 mL of nonionic iodinated contrast; 30 mL saline flush was used to eliminate contrast from the right ventricle; contrast injection rate was 3–6 mL/sec; test bolus was used to calculate the peak contrast enhancement and determine the correct scan delay. Scan range was started below the tracheal bifurcation to the diaphragm or to the lower cardiac border. Imaging was started 9 mm above the ostium of the left main coronary artery or top level of the left anterior descending artery and continued to 9 mm below the inferior aspect of the heart. The following imaging parameters were used: 100 or 120 kVp; variable mAs; 0.28 sec rotation time; collimation, 128 x 0.6 mm; HR dependent pitch; HR dependent acquisition time. Patients were scanned throughout the cardiac cycle with a retrospective electrocardiogram (ECG) gated technique using dose-modulation. Reconstruction was performed using 0.75 mm slice thickness, 0.5 mm reconstruction spacing. To determine the phase of RR interval used for CAD analysis, variable phase was used depending on HR and motion artifact; most coronary artery segments were evaluated between 70–80% RR interval. Iterative reconstruction was performed with B26f ASA/I41f kernels. Curved multiplanar reformations, ventricular function and volume analysis, and assessment of wall motion abnormality was performed on a separate workstation using TeraRecon (Foster City, California, USA).

## Image interpretation

Each coronary artery was assessed based on the Coronary Artery Disease—Reporting and Data System (CAD-RADS) [10] by a board-certified, level II or level III trained, cardiac radiologist. Coronary artery disease (CAD) was defined as a CAD-RADS score of ≥1. Hemodynamically significant stenosis was defined as a CAD-RADS score of ≥3.

## Statistical analysis

All statistical analyses were performed with SPSS 21.0 (IBM, Armonk, NY, USA). Group comparisons were performed using Chi-square test for categorical variables and Mann-Whitney for continuous variables. Logistic regression was used to measure the strength of association of CAD and potential risk factors. Risk factors included in the logistic regression included other clinical variables determined to have a possible association with CAD on CCTA on univariate analysis: age, diabetes and hypertension. All tests were two-tailed and a p-value of 0.05 was selected for statistical significance.

## Results

The 1,420 patients included in this study had demographic and medical characteristics as displayed in Table 1. Median age was 38 (range 18–50) with 50% female patients. A total of 201 (14.2%; 95% CI 12.44–16.07%) patients had coronary artery disease on the CCTA (CAD-RADS ≥ 1).

A total of 146 (10.3%) of the 1,420 included patients were identified as marijuana users. On average, marijuana users were younger and more frequently male compared with nonusers.

**Table 1. Demographic information.**

| | ALL (N = 1420) | Female (N = 711) | Male (N = 709) | p-value |
|---|---|---|---|---|
| **Continuous variables** | **median (IQR)** | **median (IQR)** | **median (IQR)** | |
| *Age (years)* | 38 (33–44) | 46 | 35 (30–38) | Not tested* |
| *Framingham score (%)* | .90 (.3–2.29) | 0.50 (0.20–1.40) | 1.40 (.50–3.95) | <0.001 |
| *Triglycerides (mg/dL)* | 120 (84–190) | 112 (79–166) | 129 (92–214.75) | <0.001 |
| *Total Cholesterol (mg/dL)* | 173 (149–201) | 173 (151–200) | 173 (148–202) | 0.619 |
| *HDL (mg/dL)†* | 44 (35–54) | 47 (39–59) | 40.5(32–49) | <0.001 |
| *LDL (mg/dL)‡* | 97 (77–121) | 97.5 (76–120) | 97 (78.5–122) | 0.64 |
| **Categorical variables** | **n (%)** | **n (%)** | **n (%)** | |
| *Hypertension* | 543 (38) | 307 (43) | 236 (33) | <0.001 |
| *Diabetes* | 190 (13) | 122 (17) | 68 (10) | <0.001 |
| *Previous Stroke* | 46 (3.2) | 35 (4.9) | 11 (1.6) | <0.001 |
| *Anxiety* | 260 (18) | 159 (22) | 101 (14) | <0.001 |
| *Tobacco Smoking* | 512 (36) | 196 (28) | 316 (45) | <0.001 |
| *Cocaine Recent* | 57 (4) | 12 (2) | 45 (6) | <0.001 |
| *Cocaine Remote* | 72 (5) | 16 (2) | 56 (8) | <0.001 |
| *Marijuana Use* | 146 (10) | 33 (5) | 113 (16) | <0.001 |
| *Obesity (> 30 BMI)* | 749 (53) | 421 (60) | 328 (47) | <0.001 |
| *CAD-RADS ≥ 1* | 201 (14.2) | 100 (14.1) | 101 (14.2) | 0.922 |

*Not tested: age distinction for men and women made by inclusion criteria.

†: High-Density Lipoprotein

‡: Low-Density Lipoprotein

They had a higher frequency of tobacco smoking and a lower frequency of hypertension and diabetes. There was no statistical difference in lipid profile values between marijuana users and nonusers (Table 2).

Table 3 outlines the CAD-RADS scores among marijuana users versus nonusers. A total of 10 (6.8%) of the 146 marijuana users as compared to 191 (15.0%) of the 1274 nonusers had

**Table 2. Selected demographic information comparison: Marijuana users vs nonusers.**

| Risk Factor | Marijuana Users (N = 146) | Marijuana Nonusers (N = 1274) | p-value |
|---|---|---|---|
| **Continuous variables** | **median (IQR*)** | **median (IQR)** | |
| *Age, years* | 36 (29–38) | 40 (34–34) | <0.001 |
| *Triglycerides (mg/dL)* | 113 (85.25–191.75) | 118 (83–188) | 0.964 |
| *Total Cholesterol (mg/dL)* | 162 (136.25–202) | 173 (151–201) | 0.083 |
| *HDL † (mg/dL)* | 42 (34–51.5) | 45 (35–54) | 0.212 |
| *LDL ‡ (mg/dL)* | 96 (66.5–120.5) | 97 (78–121) | 0.210 |
| **Categorical variables** | **n (%)** | **n (%)** | |
| *Sex, Male* | 113 (77) | 596 (47) | <0.001 |
| *Hypertension* | 40 (27) | 503 (40) | 0.004 |
| *Diabetes* | 8 (6) | 182 (14) | 0.003 |
| *Tobacco Smoking* | 108 (74) | 404 (32) | <0.001 |
| *Obesity (> 30 BMI)* | 55 (38) | 694 (55) | <0.001 |

*: interquartile range

†: High-Density Lipoprotein

‡: Low-Density Lipoprotein

**Table 3. Results of CCTA\* among marijuana users vs nonusers.**

| CAD-RADS SCORE † | Marijuana Users n (%) | Marijuana Nonusers n (%) |
|---|---|---|
| *0* | 136 (93.2) | 1080 (84.8) |
| *1* | 5 (3.4) | 100 (7.8) |
| *2* | 3 (2.1) | 55 (4.3) |
| *3* | 1 (0.7) | 20 (1.6) |
| *4* | 1 (0.7) | 16 (1.3) |
| *5* | 0 (0) | 0 (0) |

CAD-RADS score of 3 or greater corresponds to stenosis of 50% or greater in any of the major coronary vessels, and may represent hemodynamically-significant CAD

\*: Coronary CT angiography

†: Coronary Artery Disease—Reporting and Data System

CAD on CCTA (p = 0.008). The negative association between marijuana use and premature CAD disappeared after accounting for known CAD risk factors including advanced age, diabetes, and hypertension in a logistic regression analysis (SE = 0.344; 95% CI 0.299–1.15; p = 0.12, see Table 4).

A total of 36 (18.8%) of the 191 nonusers with CAD had hemodynamically significant disease on CCTA. This ratio is comparable in marijuana users with 2 (20%) of the 10 marijuana users with CAD showing evidence of hemodynamically significant disease on CCTA. One marijuana user had an estimated 70% stenosis in the proximal left anterior descending artery on CCTA (CAD RADS score = 4). This patient was the only marijuana user to undergo further cardiac testing. Exercise stress testing performed within 24 hours of CCTA did not induce any hemodynamic instability or obvious ischemic changes on ECG. Myocardial perfusion single-photon emission computed tomography (SPECT) showed no inducible ischemia or prior myocardial infarction (MI). No conventional coronary angiography was performed on this patient or any of the other patients using marijuana during the study period.

## Discussion

The results demonstrate a relatively low frequency of CAD in a younger, marijuana-using patient subgroup. A study evaluating this same young to middle age patient cohort published by our group in 2018 determined that advanced age, hypertension, diabetes and elevated triglycerides were the predominant risk factors for the development of premature CAD [9]. On the contrary, this study demonstrates that marijuana use may have a neutral effect against the development of CAD in young to middle aged adult patients.

Basic science research has identified that tetrahydrocannabinol, the active compound in marijuana, activates both CB1 and CB2 receptors. Activation of CB1 receptors has been shown

**Table 4. Multivariate analysis of CAD risk factors.**

| | | B | Standard Error | Wald | df | P-Value | Exp(B) | 95% CI for EXP(B) | |
|---|---|---|---|---|---|---|---|---|---|
| | | | | | | | | Lower | Upper |
| *Step 1\** | **Age** | 0.065 | 0.012 | 29.908 | 1 | 0.000 | 1.067 | 1.042 | 1.092 |
| | **Diabetes** | 0.543 | 0.202 | 7.233 | 1 | 0.007 | 1.720 | 1.159 | 2.555 |
| | **Hypertension** | 0.259 | 0.165 | 2.477 | 1 | 0.115 | 1.296 | 0.938 | 1.789 |
| | **Marijuana Use** | - 0.534 | 0.344 | 2.414 | | 0.12 | 0.586 | 0.299 | 1.15 |

\*: Variable(s) entered on step 1: Marijuana

to be pro-atherogenic as a result of increased inflammation, whereas activation of CB2 receptors has been shown to be anti-atherogenic as a result of decreased inflammation [11]. Thus, the effects of marijuana on atherosclerosis may be organ-specific depending on the relative concentration of CB1 to CB2 receptors. These results indicate that marijuana is not likely to be pro-atherogenic with respect to the coronary arteries. Additional basic science research in this area is necessary to further explore this relationship.

The Coronary Artery Risk Development in Young Adults (CARDIA) study, a longitudinal cohort study that included 4286 marijuana users, demonstrated that cumulative lifetime and recent marijuana use had no association with CAD, corresponding with our findings [12]. Additionally, the CARDIA study also demonstrated that in patients with a history of MI, smoking marijuana increases the risk of recurrent MI. Exclusion criteria in our study excluded patients with history of MI, so direct comparison of this finding with our study cohort cannot be made.

Marijuana can increase sympathetic nervous system activity causing an increase in heart rate. It can also increase systolic and diastolic blood pressures and may cause coronary vasospasm [13–16]. It is by these mechanisms whereby marijuana is postulated to increase myocardial oxygen demand and potentially increase a patient's risk of acute coronary syndrome (ACS) or anginal chest pain. Our study did not evaluate these relationships and was limited to determination of CAD risk in younger patients. Further studies are required to determine the association, if any, between marijuana and ACS.

Overall, we noted a relatively high number of young patients presenting with chest pain and marijuana use. In addition to anxiety, sympathetic nervous system activation by marijuana may have caused the anginal symptoms among our young patients. Transient coronary vasospasm is also a potential cause as suggested by a few published case reports [14–16]. Although CCTA has excellent negative predictive value for diagnosing CAD [17], Kang et al. found poor sensitivity of CCTA for diagnosing coronary vasospasm [18]. This is another topic that requires further investigation in patients using cannabis.

Concerning usage of other drugs among this young patient cohort, previously published findings demonstrated no association between the development of premature CAD and recent cocaine use [9]. Further studies are also necessary to determine any potential association between other substances of abuse, such as methamphetamine or heroine, and premature CAD.

## Study limitations

Marijuana users were identified by either self-reporting or a positive urine toxicology screen within one year of presentation for chest pain. However, it was not possible to determine the manner of marijuana ingestion (vaporizers, cigarettes, food, oils, etc.) based on retrospective chart review. The multiple possible forms of ingestion also make it difficult to quantify the amount of marijuana used as it is not as straightforward as determining the number of pack-years for cigarette smoking. The duration of use was also unavailable to us in the charts. This did not allow for the assessment of a dose-response relationship between marijuana and CAD. Diagnostic tests that involve measurement of fractional flow reserve may be more accurate than CCTA in detection of CAD, yet these are not routinely performed in assessment of patients in the ED setting and were not available for this retrospective study.

The baseline demographic characteristics of marijuana users and nonusers varied significantly leading to potentially significant confounding variables. One possible way to account for this is propensity score matching. However, this was not possible because of the limited number of marijuana users (10.3%) compared to nonusers (89.7%) in our patient population.

## Conclusion

There is no association between marijuana use and the presence of coronary artery disease on coronary CT angiography in young to middle aged patients presenting with chest pain. Our results correlate with findings from the CARDIA study and other recent publications. Additional research is needed to evaluate potential long-term effects of cannabis with relation to coronary artery disease.

## Author Contributions

**Conceptualization:** Jeremy R. Burt, Ali M. Agha.

**Data curation:** Jeremy R. Burt, Julie Pepe.

**Formal analysis:** Jeremy R. Burt, Ali M. Agha, Julie Pepe.

**Investigation:** Jeremy R. Burt, Ali M. Agha.

**Methodology:** Jeremy R. Burt, Ali M. Agha, Julie Pepe.

**Project administration:** Jeremy R. Burt.

**Supervision:** Jeremy R. Burt.

**Validation:** Jeremy R. Burt, Basel Yacoub, Aryan Zahergivar, Julie Pepe.

**Writing – original draft:** Jeremy R. Burt.

**Writing – review & editing:** Jeremy R. Burt, Ali M. Agha, Basel Yacoub, Aryan Zahergivar.

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
