## [Decision Letter · Decision Letter 0]

12 Dec 2019

PONE-D-19-31969

Marijuana Use and Coronary Artery Disease in Young Adults

PLOS ONE

Dear Dr. Burt,

Thank you for submitting your manuscript to PLOS ONE. After careful consideration, we feel that it has merit but does not fully meet PLOS ONE’s publication criteria as it currently stands. Therefore, we invite you to submit a revised version of the manuscript that addresses the points raised during the review process.

We would appreciate receiving your revised manuscript by Jan 26 2020 11:59PM. To enhance the reproducibility of your results, we recommend that if applicable you deposit your laboratory protocols in protocols.io, where a protocol can be assigned its own identifier (DOI) such that it can be cited independently in the future. For instructions see: http://journals.plos.org/plosone/s/submission-guidelines#loc-laboratory-protocols

We look forward to receiving your revised manuscript.

Kind regards,

Corstiaan den Uil

Academic Editor

PLOS ONE

Reviewers' comments:

Reviewer's Responses to Questions

**Comments to the Author**

1. Is the manuscript technically sound, and do the data support the conclusions?

Reviewer #1: Partly

2. Has the statistical analysis been performed appropriately and rigorously? 

Reviewer #1: I Don't Know

3. Have the authors made all data underlying the findings in their manuscript fully available?

Reviewer #1: Yes

4. Is the manuscript presented in an intelligible fashion and written in standard English?

Reviewer #1: Yes

5. Review Comments to the Author

Reviewer #1: Overall, a well written and interesting article. Few points that concern me regarding the whole overarching way that CAD was evaluated in this population - namely using coronary CT.

- if you read any guideline outlining evaluation of chest pain in the ED, it does NOT usually jump straight to coronary CT (ex: https://www.uptodate.com/contents/evaluation-of-emergency-department-patients-with-chest-pain-at-low-or-intermediate-risk-for-acute-coronary-syndrome#H24 )

- which makes me wonder - were these CTs ordered on appropriate people in the first place? They are meant to be used as a rule-out in those of low to intermediate risk for the most part

- And if that is the case - that they were ordered on people of low risk of having CAD - isn't it expected that you would find no difference in CAD between the two groups (since they both in theory should not have much, if any?)

- Use of coronary CT to evaluate CAD without using FFR may not yield as specific of results. Traditional CAD RADS does not assess for FFR https://cdn.ymaws.com/scct.org/resource/resmgr/cad-rads/scct_jcct_cad-rads.pdf

- You should just outline the pitfalls of all of this in the article

Next, another few questions:

- power calculations for group sizes? IE are there enough marijuana users for you to say these conclusions?

- why were no troponins compared between groups?

- were these patients seen and evaluated by a cardiologist before the CT was ordered, or was this done by the ED?

Finally, a few comments about specific lines of the article.

61 - this sentence outlines two very opposite effects, so maybe this should be expanded on (ie: increased BP and hypotension)

63 - this sentence could be more clear. I also don't have access to the article in reference #2 - did they really say stroke index? Or was this supposed to be stroke volume index, or just plain stroke volume?

93 - not sure that I agree with using a one time marijuana detection on urine screening to say whether or not a person used marijuana around the time of their chest pain presentation. I think this should be commented on in the limitations section

96 - definition of hypertension is not correct, because we would never say someone has hypertension after seeing them in the ED with a SBP >140 while they were in hospital presenting with chest pain. Maybe if their SBP was much greater. This should be a limitation in study. Or potentially you could reanalyze after removing all those with those HTN definitions.

132 - interesting that there was 50% females.... which is not traditionally the same percentage of all comers with MIs

147 - requires clarification. Maybe "A total of 36/191 of the nonusers with scores >1..."

148 - I think your phrasing in this part shows bias. "Only 2/10" implies that this is a small number, though percentage wise it is (nonstastically) more than the other group.

6. PLOS authors have the option to publish the peer review history of their article (what does this mean?). If published, this will include your full peer review and any attached files.

Reviewer #1: No

---

## [Author Response · Author response to Decision Letter 0]

9 Jan 2020

Review

1. Overall, a well written and interesting article. Few points that concern me regarding the whole overarching way that CAD was evaluated in this population, namely using coronary CT. if you read any guideline outlining evaluation of chest pain in the ED, it does NOT usually jump straight to coronary CT (ex: https://www.uptodate.com/contents/evaluation-of-emergency-department-patients-with-chest-pain-at-low-or-intermediate-risk-for-acute-coronary-syndrome#H24 ). Which makes me wonder - were these CTs ordered on appropriate people in the first place? They are meant to be used as a rule-out in those of low to intermediate risk for the most part

• CCTA was not the initial diagnostic test performed for patients presenting to the ED with chest pain in this study. It was performed where deemed appropriate by the ED physician or the cardiologist consulted. The joint guidelines by American College of Radiology Appropriateness Criteria Committee and the American College of Cardiology Appropriate Use Criteria Task Force list CCTA as an appropriate diagnostic modality for both patients with equivocal initial diagnosis of NSTEMI/ACS and those with low/intermediate likelihood initial diagnosis of NSTEMI/ACS.

https://www.jacr.org/article/S1546-1440(15)00680-8/pdf

2. If that is the case - that they were ordered on people of low risk of having CAD - isn't it expected that you would find no difference in CAD between the two groups (since they both in theory should not have much, if any?)

• A major strength of cCTA is its high sensitivity and negative predictive value. As such, we believe it would be an appropriate modality to detect CAD if present in low or intermediate risk patients.

3. Use of coronary CT to evaluate CAD without using FFR may not yield as specific of results. Traditional CAD RADS does not assess for FFR https://cdn.ymaws.com/scct.org/resource/resmgr/cad-rads/scct_jcct_cad-rads.pdf .You should just outline the pitfalls of all of this in the article

• This has been addressed in the section on study limitations.

Questions

4. Power calculations for group sizes? IE are there enough marijuana users for you to say these conclusions?

• We initially expected to see a difference in incidence of CAD between marijuana users and non-users.

• The following parameters were used to calculate our sample size: percentage of patient population who use marijuana (10%); incidence of CAD in marijuana users (10 %); incidence of CAD in non-users (20%); alpha (0.05); power (0.8).

• The estimated sample size for non-users: 1171; users: 117. The population size of our study was non-users: 1274; users: 146. This leads us to believe that our study was appropriately powered to detect statistically significant difference between the groups, if such difference exists.

5. Why were no troponins compared between groups?

• The main goal of this study was to detect frequency of CAD on CCTA among users and non-users of marijuana presenting with chest pain. As we were not assessing frequency of ischemia and MI in our population, troponin levels were not compared between the groups.

6. Were these patients seen and evaluated by a cardiologist before the CT was ordered, or was this done by the ED?

• The guidelines for appropriate utilization of cardiovascular imaging, published jointly by ACR and ACC, do not necessitate assessment by a cardiologist before the CCTA is ordered for a patient in the ED. Yet at our university hospital, a cardiologist is often consulted in the ED before CCTA is ordered.

Comments

7. Line 61: this sentence outlines two very opposite effects, so maybe this should be expanded on (ie: increased BP and hypotension)

• Marijuana causes an increase in both systolic and diastolic blood pressures in supine patients. However it may also induce orthostatic hypotension and orthostatic presyncope. This has been reported in a 2016 review article by Franz et al. titled “Marijuana Use and Cardiovascular Disease” and published in Cardiology in Review. This line has been rephrased in the article to better illustrate this finding.

8. Line 63: this sentence could be more clear. I also don't have access to the article in reference #2 - did they really say stroke index? Or was this supposed to be stroke volume index, or just plain stroke volume?

• The following is quoted directly from the referenced article:

“Sonographic evaluations demonstrate that cannabis use does not change end systolic volume or cardiac index, but does decrease end diastolic volume, stroke index, and ejection fraction. These findings suggest that the cardiac effect of marijuana is mainly chronotropic; in addition, the cardiac index does not change because the increased heart rate is coupled with a decreased stroke index.”

• Stroke index, as used in the referenced review, refers to stroke volume index which is also reflective of the stroke volume. The term used has been changed to stroke volume in order to avoid any confusion. 

9. Line 93: not sure that I agree with using a one time marijuana detection on urine screening to say whether or not a person used marijuana around the time of their chest pain presentation. I think this should be commented on in the limitations section

• The purpose of this study is to determine the frequency of CAD among young adults presenting with chest pain who currently use marijuana as compared to nonusers. We aimed to capture a patient population which uses marijuana habitually. The criteria used were either self-reported marijuana use or positive urine toxicology screening for marijuana during this encounter or a previous encounter within 1 year of the ED visit. Our aim was not to only capture patients who used marijuana around their chest pain presentation.

10. Line 96: definition of hypertension is not correct, because we would never say someone has hypertension after seeing them in the ED with a SBP >140 while they were in hospital presenting with chest pain. Maybe if their SBP was much greater. This should be a limitation in study. Or potentially you could reanalyze after removing all those with those HTN definitions.

• The blood pressure measurements that were used were taken from a patient’s last clinic visit or inpatient stay and not during the current ED visit. This has been clarified in the manuscript.

11. Line 132: interesting that there was 50% females.... which is not traditionally the same percentage of all comers with MIs

• It is worthy to point out that the inclusion age range was wider for women than for men. Line 88: “The search was limited to patients between the ages of 18 and 40 for men, and 18 and 50 for women.” This may have led to an almost 50% female patient population in our study as opposed to the male predominant chest pain patient population

12. Line 147: requires clarification. Maybe "A total of 36/191 of the nonusers with scores >1..."

• This sentence has been rephrased to clarify its meaning

13. Line 148: I think your phrasing in this part shows bias. "Only 2/10" implies that this is a small number, though percentage wise it is (nonstastically) more than the other group.

• This sentence has been rephrased to clarify its meaning

---

## [Editor Report · Decision Letter 1]

14 Jan 2020

Marijuana Use and Coronary Artery Disease in Young Adults

PONE-D-19-31969R1

Dear Dr. Burt,

We are pleased to inform you that your manuscript has been judged scientifically suitable for publication and will be formally accepted for publication once it complies with all outstanding technical requirements.

With kind regards,

Corstiaan den Uil

Academic Editor

PLOS ONE
---

## [Editor Report · Acceptance letter]

16 Jan 2020

PONE-D-19-31969R1 

Marijuana Use and Coronary Artery Disease in Young Adults 

Dear Dr. Burt:

I am pleased to inform you that your manuscript has been deemed suitable for publication in PLOS ONE. Congratulations! Your manuscript is now with our production department. 

With kind regards,

on behalf of

Dr. Corstiaan den Uil 

Academic Editor

PLOS ONE